# Group-Living Spider *Cyrtophora citricola* as a Potential Novel Biological Control Agent of the Tomato Pest *Tuta absoluta*

**DOI:** 10.3390/insects14010034

**Published:** 2022-12-30

**Authors:** Thomas A. Roberts-McEwen, Ella K. Deutsch, Monica A. Mowery, Lena Grinsted

**Affiliations:** 1School of Biological Sciences, University of Portsmouth, King Henry Building, King Henry 1 Street, Portsmouth PO1 2DY, UK; 2School of Life Sciences, University of Nottingham, University Park, Nottingham NG7 2RD, UK; 3Jacob Blaustein Institutes for Desert Research, Mitrani Department of Desert Ecology, Ben-Gurion University of the Negev, Midreshet Ben-Gurion 8499000, Israel

**Keywords:** sociality, communal, colonial spiders, predator-prey, food security, pesticide resistance, sustainable agriculture

## Abstract

**Simple Summary:**

The tomato leafminer, *Tuta absoluta*, is a devastating pest moth of commercially important crops like tomato and potato. This moth has developed resistance to insecticides; therefore, novel approaches, like using natural predators, are needed to combat infestations. We explored the use of tropical tent web spiders, *Cyrtophora citricola*, as biological control agents, as these spiders live in groups and are not cannibalistic, and thus, create large, predator-dense webs. Furthermore, their global range overlaps with regions of moth infestations. In lab settings, we introduced different prey types to small colonies of spiders of varying body sizes and found that spiders were equally efficient at capturing pest moths and easily-caught fruit flies (*Drosophila hydei*). Larger spiders built larger webs and were better at catching prey. Spiders from southern Spain were large enough to capture pest moths during the tomato growing season, but >50% of spider egg sacs were attacked by egg predatory wasps (*Philolema palanichamyi*). *Cyrtophora citricola* spiders, therefore, have the potential to be an effective biological control agent of flying insect pests, at least after growing to medium-sized juveniles, and if wasp infections are controlled, forming part of integrated pest management to defend against pest infestations in the future.

**Abstract:**

Group-living spiders may be uniquely suited for controlling flying insect pests, as their high tolerance for conspecifics and low levels of cannibalism result in large, predator dense capture webs. In laboratory settings, we tested the ability of the facultatively communal spider, *Cyrtophora citricola*, to control the tomato leafminer, *Tuta absoluta*; a major pest of tomato crops worldwide. We tested whether prey capture success was affected by spider body size, and whether prey capture differed among *T. absoluta*, flightless fruit flies (*Drosophila hydei*), and larger black soldier flies (*Hermetia illucens*). We found that larger spiders generally caught more prey, and that prey capture success was similar for *T. absoluta* and easily caught fruit flies, while black soldier flies were rarely caught. We further investigated the seasonal variations in web sizes in southern Spain, and found that pest control would be most effective in the tomato planting and growing season. Finally, we show that *C. citricola* in Spain have >50% infection rates of an egg predatory wasp, *Philolema palanichamyi*, which may need controlling to maintain pest control efficacy. These results suggest that using *C. citricola* as a biological control agent in an integrated pest management system could potentially facilitate a reduction of pesticide reliance in the future.

## 1. Introduction

Climate change due to human overpopulation and fossil fuel dependence is facilitating the spread of invasive pest species of agricultural crops by expanding their habitable environment ranges [1]. Increasing interconnectedness of the global food chain also allows for the anthropogenic introduction of potentially devastating agricultural pests, increasing pesticide reliance worldwide [1]. The tomato leafminer, *Tuta absoluta* (Meyrick, 1917) (Lepidoptera: Gelechiidae), is a species that has undergone rapid range expansion, reaching near-global ubiquity [2,3,4]. *Tuta absoluta* is a species of neotropical, oligophagous moths of solanaceous crops, with a preference for tomato [5]. The species also associates with a number of other host plants, many of which are agricultural crops, such as potatoes, bell and chilli peppers, and aubergine [5]. Crop damage is caused by larval feeding, which affects all epigeal plant parts, most notably the leaves [5,6]. Larvae burrow to consume the mesophyll layer, reducing photosynthetic surface area and resulting in diminished plant growth and fruit yield (Figure 1a) [6]. Larvae can also directly attack the tomato fruit, causing aesthetic damage and rendering the crop unmarketable [4]. Larvae live within the leaf until they pupate, making them difficult to control during juvenile life stages, due to predators lacking the capability to effectively target them within the leaf [6,7]. Here, we explore the potential for the use of the group-living tropical tent web spider, *Cyrtophora citricola* (Forskål, 1775) (Araneae: Araneidae) as a biological control agent of the adult, flying moth. The geographical distributions of *C. citricola* and *T. absoluta* overlap in large parts of the world, making this web-building spider a suitable candidate for biological pest control.

*Tuta absoluta* has infested 60% of global tomato-cultivated land [2], and can cause 80–100% yield reduction in both open-field and protected cultivations if left untreated [5]. As tomato is among the most cultivated and consumed vegetable crops worldwide [5], finding suitable control strategies to reduce the ubiquity and feeding voracity of *T. absoluta* is of increasing importance [3,8]. *Tuta absoluta* infestations are causing yield loss and, therefore, economic detriment that disproportionately affects low- and lower-middle-income countries (LMICs) [9,10]. Chemical insecticides have historically been favoured as mitigators of localised damage caused by numerous phytophagous insect species [11]. However *T. absoluta* has developed resistance to many commonly used synthetic insecticides, ultimately resulting in difficulty controlling infestations [2,4,12,13]. Furthermore, pesticide dependency in LMICs has resulted in a wide range of detrimental outcomes for both human and environmental health [10]. As the workforce in LMICs moves away from farming and toward industrialised society, domestic food is increasingly produced by fewer, often educationally disadvantaged individuals, and LMICs with long growing seasons resort to increasing non-traditional crop export to temperate zones to earn valuable foreign currency [10,14]. These socio-economic and agricultural shifts are not currently possible without increased crop yield, facilitated by the use of chemical pesticides, many of which are illicit, homemade mixes that are sold more affordably than those that are regulated [14]. Furthermore, these pesticides are often used by farmers with little means of procuring appropriate personal protective equipment (PPE), or little power or willingness to ensure its use within the workforce [14], resulting in frequent incidences of poisoning [15,16].

Currently, chemical insecticides remain the most widely used method of controlling *T. absoluta*, despite their inefficiency and danger to humans and the environment [13]. It is, therefore, important to explore whether natural predators of agricultural pest species can be used to negate the detrimental effects of herbivore infestation, with the aim of reducing the reliance on mass distribution of toxic chemical pesticides [15,16]. Research into finding appropriate biological control agents against *T. absoluta* is ongoing. Due to the ubiquity of the pest, it is likely that multiple biological control species will be required to meet the needs of the diverse ecological systems and climates it has invaded. Various approaches to *T. absoluta* population management have been tested, including the use of predators, parasitoids, and entomopathogens [17]. It is likely that a combination of these methods will prove most effective as components of an integrated pest-management system [18]; however, the search for a highly effective combination of natural predators is ongoing [4]. Furthermore, most current biological control methods for *T. absoluta* infestation rely on controlling the pest at its larval stage [19]. Methods currently used for targeting exclusively larval *T. absoluta* instars include, but are not limited to: the use of bacterial toxins produced by *Bacillus thuringiensis* (Berliner, 1915) (Bacillales: Bacillaceae) [20]; granulovirus isolates from *Phthorimaea operculella* (Zeller, 1873) (Lepidoptera: Gelechiidae) that delays *T. absoluta* larval growth and reduces pupation [21]; larval parasitoids such as *Dolichogenidea (=Apanteles) gelechiidivoris* (Marsh, 1975) (Hymenoptera: Braconidae) [22]; entomopathogenic fungi, such as *Beauveria bassiana* (Bals.-Criv., Vuill., 1912) (Hypocreales: Cordycipitaceae) and *Metarhizium anisopliae* (Metschn., Sorokīn, 1883) (Ascomycota: Hypocreales) [23]; and entomopathogenic nematodes, such as *Heterorhabditis bacteriophora* (Poinar, 1976) (Nematoda: Heterorhabditidae) and *Steinernema carpocapsae* (Weiser, 1955) (Rhabditida: Steinernematidae) [24]. Due to the short life cycle of *T. absoluta*, as well as their overlapping generations [2], simultaneous removal of individuals at all instars must be achieved to produce an integrated pest-management approach effective enough to prevent *T. absoluta* reinfestation.

Web-building spiders are key predators of flying insects [18]. One particularly promising yet unexplored potential biological control agent of the adult instar of *T. absoluta* is *C. citricola*, a species of facultatively group-living orb-weaving spider (Figure 1b) [25,26]. *Cyrtophora citricola* occurs in Mediterranean Europe, Africa, Asia, and the Middle East [25,27], all of which are regions that contain LMICs suffering from *T. absoluta* invasion [10,19,28], highlighting the geographical suitability of *C. citricola* as a biological control agent of *T. absoluta*. Furthermore, the group-living tendencies of *C. citricola*, with their high levels of conspecific tolerance and low levels of cannibalism, can result in high densities of predators [26,29,30]. *Cyrtophora citricola* produce non-adhesive, horizontal sheet webs, which are defended territorially from conspecifics despite high tolerance of colony members occupying adjacent webs (Figure 1c,e) [25,31,32]. Spiders commonly attach their individual webs together to form large colonies, and the connecting threads in *C. citricola* colonies are used communally (Figure 1c,e) [33,34]. Colony geometry facilitates the exploitation of the ‘ricochet effect’, wherein prey items that escape the web of one individual may fall into the web of an adjacent conspecific in the colony [35]. The importance of the unique construction of the three-dimensional capture web structure for enhanced prey capture capability is further described in Su and Buehler (2020), who suggest that connecting threads also play a role in filtering out prey of low impact velocity, and protecting the individual residing at the centre of the web [36]. Two major components that contribute to the robustness and fast repairability of *C. citricola* webs are: (a) that web silk displays non-linear behaviour, wherein it may soften or stiffen at a molecular level as a response to strain, and (b), that tension in the main load-bearing strand can be released to multiple other strands in the event of it breaking, preventing web collapse [36]. Therefore, in the event of damage being done to *C. citricola* web structures, biological control capability can be effectively maintained. Their potential for effective use as biological control agents is therefore greater than that of more aggressive, solitary spiders that are prone to cannibalism [37,38,39]. Past studies on the use of spiders for pest control predominantly focus on non-web-building, solitary species, due to their propensity to capture prey from the crop surface, as well as their ability to consume less motile prey arthropod instars [7,40,41]. However, spiders that can form groups of hundreds, or even thousands, of interconnected webs can provide large surface areas of capture webs capable of intercepting high frequencies of airborne arthropods [37,42]. *Cyrtophora citricola* colonies also provide a substrate for other spider species, such as kleptoparasitic *Argyrodes* spp. (Araneae: Theridiidae), and *Holocnemus pluchei* (Araneae: Pholcidae) [33], further increasing predator density and, therefore, potentially increasing pest insect capture capability within colonies.

In this study, we ask whether *C. citricola* has the potential to act as a biological control agent of *T. absoluta*. We use southern Spain as a case study, where *C. citricola* has a strong association with prickly pear cactus (*Opuntia* spp.) that has historically been used as fences and field borders around agricultural fields [42,43]. Therefore, spider colonies are commonly found in association with field-grown crops in this region, with a potential for providing biological pest control [44]. First, we record the capture rate of *T. absoluta* by small spider colonies in lab settings, and test the effect of spider body size (and therefore spider web size) on capture success. We compare the capture rates of *T. absoluta* with that of easily caught flightless fruit flies (*Drosophila hydei*) as a control. Next, we investigate the seasonal variations in spider web sizes in southern Spain and relate that, and the potential for *T. absoluta* control, to the tomato-growing season. Finally, we consider possible inhibitors of *C. citricola* prey capture efficiency, with a focus on the egg predator *Philolema palanichamyi* (Narendran, 1984) (Hymenoptera: Eurytomidae).

This paper aims to: (a) compare the ability of *C. citricola* to capture three different prey items: the tomato leafminer, *T. absoluta*; the easily-caught and similarly sized flightless fruit flies, *D. hydei*; and the much larger black soldier flies, *Hermetia illucens*; and whether a prey type preference was exhibited; (b) test the effect of *C. citricola* body size on prey capture capability and prey size preference; (c) assess whether seasonal changes in web size may affect prey capture efficacy and, therefore, biological control potential of *C. citricola* in southern Spain; and (d) estimate the potential for egg predatory wasps, *P. palanichamyi*, to negatively affect *C. citricola* pest control capability. Spiders used in the laboratory-based experiments were reared from eggs in the laboratory, whereas seasonal web size and wasp infection data were collected from wild populations.

## 2. Materials and Methods

### 2.1. Collecting and Rearing C. citricola Spiderlings Prior to Experimental Setup

Egg sacs of *C. citricola* (N = 87) were collected in May 2021 from four sites in southern Spain, near Cádiz, coded PA (36°40′20.39″ N, 6°23′25.91″ W), CM (36°39′25.07″ N, 6°22′19.23″ W), MA (36°39′49.41″ N, 6° 5′54.14″ W), and AQ (36°37′25.00″ N, 6°11′42.96″ W). *Cyrtophora citricola* construct a string of multiple egg sacs (Figure 1b) [45]. Only the most recently constructed egg sacs (the bottom egg sacs on each string) were collected, to prevent spiderlings emerging in transit to the UK (Figure 1b). Egg sacs were reared at 22 °C in the laboratory at the University of Portsmouth in 40 mL plastic collection tubes with foam bungs (Figure 1d). Tubes were sprayed with a very fine mist of water three times per week.

After emerging and creating capture webs within the 40 mL falcon tubes, each clutch of spiderlings was fed with five to eight D. hydei once per week. As spiderlings grew, they were transferred to either: (a) one litre (1 L) plastic tubs (diameter: 12 cm, H: 15 cm) with mesh fabric lids and web supports made from 10 cm tall rectangular wire (H:10 × L:30 cm) that was rolled up along its length to provide a structure for webs of varying sizes; or (b) large 90 cm tall (W:60 × D:60 × H:90 cm) mesh enclosures with rolled up 70 cm tall (H:70 × L:100 cm) rectangular wire as web supports (Figure 1f). Food supply was increased to ten to fifteen *D. hydei* per clutch per week as spiderlings grew. All spiderlings were kept at room temperature (~22 °C) during development. No spiders reached their final moult (sexual maturity) during this time, and therefore, due to time constraints, all spiders used in the study were juveniles in varying stages of pre-adult development. Prey capture assays were conducted from September to November 2021.

### 2.2. Rearing T. absoluta Moths and Flies

Tomato leaves infested with *T. absoluta* larvae were first provided by a local UK tomato grower (N_larvae_~50). Later, as our experiment continued past the UK tomato growing season, additional *T. absoluta* pupae (N_pupae_~100) were provided by the Centre for Agriculture and Bioscience International (CABI), Ghana. Moths from both the UK and Ghana were randomly allocated amongst experimental spider colonies, ensuring no correlation between spider body sizes and moth origin. *Tuta absoluta* larvae require fresh tomato leaves for completion of their life cycle, thus, galleries containing the larvae were carefully cut from the infected leaves and placed on lab-grown tomato plants. Most larvae successfully burrowed into the mesophyll layer of the living tomato plants (Figure 1a). The larvae and tomato plants were contained in mesh enclosures of varying sizes (small: W:30.5 × D:30.5 × H:30.5 cm; medium: W:40 × D:40 × H:60 cm; large: W:60 × D:60 × H:90 cm (Figure 1f)) depending on the size of the tomato plants, with mesh holes roughly 0.5 mm in diameter. Double-sided adhesive tape was applied to the table surface surrounding each enclosure to ensure that any potential escaping larvae would be caught by sticking onto the tape.

The larvae were left in the enclosures with tomato plants to pupate until emerging as adults, when they were removed and transferred to the spider enclosures for the prey capture assays (see below). Moths were carefully caught in 40 mL tubes within the enclosure to prevent escape. The larvae and tomato plants were sprayed with water three times per week, and the tomato plants were watered once per week. After moth rearing was complete, all enclosures and plants were frozen at −6 °C for four weeks to prevent any remaining *T. absoluta* larvae, pupae, or adults from surviving.

Black soldier flies (*H. illucens*) and fruit flies (*D. hydei*) were purchased from an online pet food supplier (Livefood UK Ltd., Axbridge, UK). The *H. illucens* were received as larvae, and were transferred to small square mesh enclosures (W:30.5 × D:30.5 × H:30.5 cm) (Figure 1f) until they eclosed. The *D. hydei* were also received as larvae, and were kept in their original cultures to eclose. All flies were kept at around 22 °C in the laboratory. If too many flies were eclosing at once, and would die before being used as prey for the spiders, the culture was refrigerated at 4–5 °C to slow development.

### 2.3. Experimental Setup of Prey Capture Assays

We created twenty experimental spider colonies, each with a colony size of five spiders. Experimental colonies were created using spiders from 13 egg sacs from the PA site only, as this site yielded the most spiderlings. The spiderlings had grown at different rates in the lab, and we selected a total of 100 juvenile spiders of as broad a range of body sizes as possible, weighing each of them to the nearest 0.1 mg using a Sartorius B120S scale. We then temporarily placed them individually in 40 mL tubes (Figure 1d) and ranked them according to body mass (spiders ranged from 0.1 mg to 52.6 mg with a mean of 5.3 mg). Next, we placed them all into twenty colonies named A to T, each colony containing five similar-sized spiders. Due to random variations in growth rates amongst juveniles, spiders had reached different juvenile instars. The smallest five individuals comprised colony A, and the largest five individuals comprised colony T. Only female spiders were used because females are the large and communal sex in this species [37,42]. Females were identified by their lighter colouration and smaller pedipalps in comparison to male counterparts. Any juveniles that could not be sexed were excluded.

Colonies A–J were established in small mesh enclosures (W:30.5 × D:30.5 × H:30.5 cm), while Colonies K–T were established in medium mesh enclosures (W:40 × D:40 × H:60 cm) to minimise the risk of cannibalism due to crowding (Figure 1f). Rolled-up wire netting provided structural support for web building (netting in small enclosures: H:20 × L:50 cm; medium enclosures: H:40 × L:70 cm) (Figure 1e,f). The enclosures were stored adjacent to laboratory windows to receive natural light, and additional electrical room lighting was provided for 8 h per day. Any spider that died during the study was replaced by another individual in the same weight bracket from the pool of spiders from the PA site and replacements were limited to one spider per colony to minimise disturbance to group composition. In total, sixteen replacements were made.

The prey capture assays were conducted from 30 September 2021 to 22 November 2021. At the end of the 6.5-week study, 84 out of 100 spiders remained. Throughout the experimental period, spiders were sprayed with water and fed three times per week.

After the end of the experiment on 26 November 2021, spider body mass, body length, and capture web diameter were measured to assess the correlations between spider size, capture web size, and prey capture success. First, each colony was sprayed with a fine mist of water to improve web visibility. Capture web sheet diameter was then measured to the nearest 1 cm with a 30 cm ruler. Next, spiders were removed from their colony and weighed individually to nearest 0.1 mg. We measured body length from the tip of the prosoma to the bottom of the abdomen to the nearest 0.01 mm using an electronic calliper [46]. To record body length, each spider was transferred to a tray and left undisturbed until becoming still, wherein the measurement was taken with little handling to mitigate stress to the animals. All spiders were then transferred into mixed colonies and kept in the laboratory under similar conditions as described above. It was not possible to follow individual spider growth from pre- to post- experiment, so we calculated average body masses, body lengths, and web sizes per experimental colony.

### 2.4. Prey Capture Assays

Four prey capture treatments were implemented: (1) a control treatment introducing five flightless fruit flies per colony to represent easy-to-catch prey that was of a similar size to *T. absoluta* (N_trials per colony_ = 5); (2) a single *T. absoluta* per colony (N_trials per colony_ = 3); (3) one flightless fruit fly together with one *T. absoluta* per colony to test for prey type preference (N_trials per colony_ = 3); and (4) one black soldier fly per colony to represent a relatively large prey item (N_trials per colony_ = 3). The choice of fly species was made partly due to their accessibility from live food retailers. Average body mass for each insect was as follows (based on weighing five live specimens per species to nearest 0.1 mg): *T. absoluta*: 1.16 mg (st.dev = 0.27 mg); *D. hydei*: 2.28 mg (st.dev = 0.19 mg); *H. illucens*: 34.24 mg (st.dev = 5.47 mg). The order of capture treatment was random with respect to colony ID, and opportunistically implemented according to when prey types became available. Flies were refrigerated for five minutes to slow their movement prior to transfer to spider enclosures.

A trial consisted of placing a single insect, or several insects, according to treatment, at the bottom of a spider enclosure, to allow prey to move about and land in spider webs of their own accord, between 10 am and 12 noon. To allow enough time for insects to intercept the spider webs, colonies were left undisturbed for the following 72 h (+/−2 h). At the end of this period, the number of insects trapped in webs was counted for each colony. All insects (both live and dead) were then removed from each enclosure and replaced with fresh ones, except in the case of *T. absoluta*, where uncaptured individuals were re-used in other trials due to short supply.

### 2.5. Seasonal Web Size Measurement and Effects of Egg Predators

Six sites around Rota, southern Spain, coded CM, PA, NN (36°39′50.38″ N, 6°22′8.56″ W); EO (36°40′35.35″ N, 6°24′6.73″ W); WP (36°40′8.13″ N, 6°23′21.48″ W); and SN (36°38′58.15″ N, 6°22′32.79″ W) were visited roughly every 6 weeks over ten months from March 2019 to January 2020 (dates: 29/03, 06/05, 09/06, 19/08, 06/10, 29/11, 24/01). At each site, between 9 and 39 m of prickly pear cactus (*Opuntia* spp.), located along field edges, were selected for seasonal observations of spider colonies. During each trip, the horizontal web sheet diameter of all individual *C. citricola* spider webs (except from very small and hard to spot hatchling webs of just a few cm) along the stretches of cactus (N_total web diameters_ = 1238) were measured to the nearest cm using a measuring tape.

Additionally, during each trip, up to three egg sac strings were collected from each of the same field sites to assess egg predator infection rates (N_total #egg sacs_ = 121). After collection, each egg sac was separated from the string, weighed, and stored in a temperature controlled room at 25 °C in falcon tubes with foam bungs. Egg sacs were misted twice weekly and monitored until spiderlings and/or *P. palanichamyi* emerged. Wasps were counted as they emerged, while photographs of spiderlings were taken for later counting due to high numbers emerging. We counted the spiderlings using the freely available software Dot Dot Goose (version 1.5.3) [47].

A further 96 egg sacs were collected from CM, AQ and SN in southern Spain in May 2022, and brought back to the lab. Here, they were kept at room temperature (~22 °C) and misted, as described above, and presence versus absence of emerging spiderlings and wasps was recorded over the following six weeks.

### 2.6. Statistics

All statistical analyses were conducted using R (version 4.1.1) [48]. The raw data is available in Appendix A.

#### 2.6.1. Spider Sizes

For each experimental colony, we calculated an average body mass, both pre- and post-experiments, as well as a colony-average body length and web size post-experiment. The distributions of all four variables were heavily left-skewed, so we used the non-parametric Spearman’s rank correlation to test the correlations between the per-colony average values of pre- and post-weights, post-weight and post-web size, post-weight and post-body length, and post-body length and post-web size, all with N = 20. We used these correlations to justify interchangeably using body mass and web size as proxies for spider body size.

We further asked whether spiders had grown over the course of the experiment by testing the difference in average body mass pre- versus post-experiment with a Wilcoxon’s test for matched pairs (N = 20). Due to the resulting significant growth of spiders over the experiment, which may have influenced their capture-abilities over time, we used the average between the pre- and post- average body masses as a response variable in the prey capture data analyses described below.

#### 2.6.2. Control Prey Capture

In the control treatment where five fruit flies were introduced per colony five times, we asked whether larger spiders generally have higher prey capture success. We did this by testing the correlation between average spider body mass and the total number of flies caught per colony (up to a max. of 25 over the five feeding trials) using a Spearman’s rank correlation test (N = 20).

#### 2.6.3. Prey Capture Treatments

We investigated the effect of prey type and spider body size on prey capture success using a Generalised Linear Model (GLM) fitted with a binomial error structure and logit link function. We created a proportional response variable in the form of binding together two vectors, one that included the number of a prey type caught per colony over three trials (and so ranging from 0 to 3) and the second that included the number of prey items not caught (=3−the number caught).

As predictor variables, we included the per-colony average spider body mass, the treatment prey type, and their interaction term. The treatment prey type had four levels, as follows: soldier flies introduced singly, *T. absoluta* introduced singly, *T. absoluta* introduced singly together with a single fruit fly, and fruit flies introduced singly together with a single *T. absoluta* (N per treatment prey type = 20; total N in the model = 80).

We checked the distribution of prey caught to ensure a lack of zero inflation, and further tested to ensure a lack of overdispersion before proceeding to significance testing. Finally, we tested to ensure the full model was significant before proceeding to test the significance of the predictor variables. Significance of the interaction term and predictor variables were tested by comparing full models with reduced models.

We further ran the full model on a subset of the dataset that excluded the black soldier fly treatment. We did this as a post-hoc test to test for any differences in prey capture success between *T. absoluta* and easily caught fruit flies.

## 3. Results

### 3.1. Spider Sizes

All non-parametric correlations between proxies for spider body size were highly significant and positive: pre- and post-experiment average body mass (rho = 0.95, *p* < 0.001) post-experiment body mass and web size (rho = 0.93, *p* < 0.001), post-experiment body mass and body length (rho = 0.98, *p* < 0.001), and body length and web size (rho = 0.86, *p* < 0.001). These strong, positive correlations justify the interchangeable use of body mass, body length, and web size as equally valid proxies for spider size. Spiders were significantly heavier after the end of the experiment (Wilcoxon’s test V = 2, *p* < 0.001); therefore, we used the average between the pre- and post-average body masses per colonies in the prey capture analyses, as described below.

### 3.2. Control Prey Capture

Larger spiders were able to capture significantly more prey in our control experiment, where multiple easily caught prey items (five wingless fruit flies) were introduced at a time (S = 58.1, *p* < 0.001, rho = 0.96, Figure 2a).

### 3.3. Prey Capture Treatments

We found a significant interaction between spider size and prey type treatment (binomial GLM, *p* = 0.0032). This means that the general increase in prey capture success for larger spiders differed according to prey type (Figure 2b,c). Prey capture of *T. absoluta* was 100% for colonies of an average body mass of ~9 mg, body length of ~5 mm and web size of ~14 cm, and above, while spiders of a range of body sizes captured 100% of fruit flies, and spiders of most body sizes tended to be unsuccessful in capturing black soldier flies (Figure 2b,c).

In a posthoc test, where black soldier flies were excluded, the interaction between spider size and prey type treatment was not significant (binomial GLM, *p* = 0.25). Instead, spider size was a highly significant predictor of prey capture success (*p* < 0.001), whereas treatment was not significant (*p* = 0.058). Hence, whilst the capture success of *T. absoluta* was slightly lower than that of flightless fruit flies, this difference was not statistically significant, suggesting that spiders had no preference for either prey type.

### 3.4. Seasonality

Naturally occurring *C. citricola* webs in southern Spain fluctuated over the year according to the breeding season: as females grew and sexually matured in spring, webs grew larger and peaked in May and June, with most webs being 20–30 cm (Figure 3). After reproducing, most adult females died and the small webs of their offspring (≤10 cm) slowly started to dominate over the summer, although a few adult, breeding females were present year round. Offspring body sizes and, hence, web sizes began to increase over autumn and winter until the next main breeding season in spring.

In southern Spain, the tomato planting and growing season (March-June) coincides with the *C. citricola* main breeding season, when webs are at their large size, while webs found during the tomato harvest season, July–October, are at their smallest size (Figure 3) [49].

### 3.5. Wasp Infection

Out of 121 collected egg sacs in 2019, 12 had to be discarded because of labelling error. Out of the remaining 109 egg sacs, 73 produced live animals with an overall infection rate of 54.8%. Of the 73 egg sacs, 33 (45.2%) produced spiderlings only, with a median of 191 hatchlings (max. = 396, average = 181.4, st.dev = 87.5). Another 37 egg sacs (50.7%) were infected with wasps and produced zero spiderlings. From these egg sacs, a median of 18 wasps emerged (max. = 79, average = 26.5, st.dev. = 21.8). In only 3 egg sacs (4.1%), did some spiderlings survive a wasp infection, and these produced both spiderlings (between 46 and 104) and wasps (between 6 and 22).

From the 96 collected egg sacs in 2022, 73 egg sacs produced live animals, with an overall infection rate of 69.9%. Out of these, the proportion of egg sacs from which some spiderlings survived an infection was 42.5% (both spiderlings and wasps emerged from 31 egg sacs), while 22 egg sacs (30.1%) produced spiderlings only and 20 (27.4%) produced wasps only.

## 4. Discussion

This study set out to test the ability of the communal spider, *C. citricola*, to capture the tomato pest, *T. absoluta*, by providing laboratory-reared *C. citricola* colonies with different prey types. Small, experimental colonies of juvenile spiders were able to capture both the leafminers and flightless fruit flies, considered to be easily caught prey, with near-equal efficiency, and spiders showed no significant preference for either species. Larger black soldier flies, however, were rarely caught in these settings. This suggests that the spiders are as likely to capture and prey on leafminers as other small, easily caught insects, and, therefore, show promise as a potentially effective biological control agent of the moth. However, spider body size, which positively correlated with web size, was a strong predictor of prey capture success of all prey types tested. The capture success of the leafminer only reached 100% when juvenile spiders were about 5 mm in body length and had webs of roughly 14 cm in diameter and above, suggesting that hatchlings and very small spiderlings would be ineffective predators of adult, flying leafminers. Our experiments were conducted in the laboratory, and the spiders were exposed to prey items for three full days, which may also not fully represent prey capture dynamics in the field. Therefore, future studies should test prey capture efficacy in field settings, where prey may have a higher chance of avoiding capture webs.

Spiders in wild colonies of *C. citricola* in southern Spain produced the largest webs in May and June, which indicates that these months are the most opportune for the use of *C. citricola* as a biological control agent. This period neatly correlates with the beginning of the tomato growing season in Andalusia, southern Spain [49], where control of *T. absoluta* is crucial for commercial tomato farms. We found that web size is a function of both spider body mass and body length in *C. citricola*, meaning that prey capture potential of a spider colony could potentially be predicted by estimating average web sizes. Measuring individual spider web sizes in field settings is quick and easy [42], and would be an undisruptive method of gauging colony-level capture rate efficacy. This could be especially useful for predicting the pest-control capability of a developing colony of biological control spiders after introduction to an agricultural system. Nevertheless, it is important to note that the spiders used in this study were juveniles, and it is, therefore, unknown whether much larger, adult individuals would expend the effort to consume small *T. absoluta* individuals, especially when larger prey is likely to be accessible to them a natural setting [42]. While further research is needed to confirm that larger spiders (subadult and adult females) will also prey on *T. absoluta*, we know from previous studies that larger spiders often catch relatively small prey, and thus, are likely to prey on the relatively small moths. In Grinsted et al. (2019), larger spiders, including adults, with webs between 20 and 37 cm in diameter preyed mainly on insects smaller than the average *T. absoluta* body length of 6 mm [50]. Indeed, median prey length for these larger females in natural field settings was 3 mm (prey body length ranged from 1–17 mm, with 75% of prey <6 mm), calculated from the raw data deposited by Grinsted et al. (2019) [42]. These results also suggest that spiders in Spain are unlikely to be effective as pest control agents during the harvest season in southern Spain [49], as spider webs are mostly too small during July–October. This further suggests that seasonal fluctuations in web sizes in a given geographical region must be taken into consideration prior to *C. citricola* application [44]. Despite the spiders’ efficacy at catching the tomato pest, it is important to consider that they are generalist predators, and are, therefore, capable of removing beneficial pollinators [7,18,44], which are crucial to the fertilisation of the tomato crop. It is, therefore, integral that the effects of biological control colonies of spiders on pollinator populations are considered in future studies.

When assessing the efficacy of a novel biological control agent, it is imperative that community ecology is considered, as interactions with other species in the community may hamper pest control abilities [29,33,38]. One possible disruptor of the effectiveness of *C. citricola* as a biological control agent is egg predation by the wasp, *P. palanichamyi*, a species that oviposits into the egg sacs of *C. citricola*, and emerging larvae consume the developing spider eggs [25]. We found an infection rate of >50% of egg sacs at our field sites near Cádiz, while Chuang et al. (2019) [25] found that about 42% of egg sac strings were infected over a larger area of southern Spain, from Cádiz to Valencia. We found large variations in spiderling survival after wasp infections, but overall, ~30–50% of sampled egg sacs produced wasps only, with zero surviving spiderlings. Hence, wasp infections may cause severe predation pressure and possibly shape extinction patterns in *C. citricola*, at least in Spain, as suggested by Chuang et al. (2019) [25]. Furthermore, by introducing high numbers of *C. citricola* to an area as a biological control agent, more egg sacs will subsequently be provided as prey for *P. palanichamyi*, resulting in population increase of the wasp, and possible local community ecology alteration [51,52]. Additionally, the implementation of additional *C. citricola* colonies into ecosystems is also likely to facilitate population expansion of both the colony-associate *H. pluchei* and kleptoparasitic *Argyrodes* spp., which could also potentially detrimentally affect *C. citricola* populations. It is, therefore, important to address how community ecology may impact *C. citricola* population dynamics and the resulting pest control efficacy [52]. Thus, we propose that species communities within spider colonies, particularly focussing on *H. pluchei* and *A. argyrodes*, as well as wasp infection rates in nearby rural populations, should be closely monitored during implementation of the biological control agent. Furthermore, in the event of wasp populations expanding and causing potential harm to both natural and biological control spider populations [52], a plan to control wasp infections must be devised. A possible future avenue for research is therefore testing the rate of increase in *P. palanichamyi* populations over multiple generations, in the presence of increasing egg sac numbers.

The proposed efficacy of *C. citricola* as a biological control agent is based on two useful facets of the species. Firstly, the evolution of group living and high conspecific tolerance confers reduced aggression toward neighbouring spiders, and therefore, fewer incidences of intraspecific attack and cannibalism as compared to solitary spiders [32,37,42]. This is likely to result in high predator density when used as a biological control agent and the ability to intercept large numbers of pest arthropods with their interconnected capture webs [37,42]. Furthermore, few studies have tested the efficacy of group-living spiders as biological control agents, many focussing on comparing web-building and non-web-building spiders, despite the aforementioned advantages of the use of communal species [44,52,53]. Secondly, the global ubiquity of *C. citricola* may result in its potential use in multiple locations worldwide, including LMICs such as those in Mediterranean Europe, Africa, Asia, and the Middle East, whose environmental health and economic stability could greatly benefit from this sustainable agricultural approach [25,27]. However, incidences of the spider becoming an agricultural pest have occurred in countries including the Dominican Republic and Columbia, where it was introduced in the last 25 years [54]. Here, colonies can comprise a great number of individuals, and excessive colony expansion results in increased capture web construction, which has been reported to asphyxiate crop plants, resulting in reduced crop yield and potentially causing economic deficit [25,54,55]. It is, therefore, important to ensure that appropriate substrate is provided for spiders to build their colonies on, and that spiders are discouraged from building webs directly on the crops, before commercial use can commence [7]. Such substrates could be natural supports, such as the *Opuntia* spp. cactus, which is both a favoured host of wild *C. citricola* colonies, and a commonly grown border plant around agricultural fields in southern Spain [43]. Inorganic frames could also be used; however, these may be less effective than preferred host plants, such as *Opuntia* cacti, as modified cactus stems provide wind protection that increases prey vibration sensitivity [43] and are, therefore, likely to benefit prey capture rate in a way that wire supports could not. In cases where introduction of *Opuntia* field borders is not possible, suitable inorganic substrates must be developed to ensure that *C. citricola* pest control colonies can reach their highest potential prey capture efficiency.

In this study, we noted multiple instances of *T. absoluta* sitting uncaptured in *C. citricola* webs in all colonies during interim checks during the three-day capture period, either suggesting that the moth is not always detectable by the spiders, or that the moths can avoid becoming caught in capture webs. It may be possible that the low body mass of *T. absoluta* results in the production of few vibratory signals, causing inconsistency in prey capture capability. In addition, moth scales, which are likely to be lost as they brush against the non-adhesive silk strands of *C. citricola* webbing, may allow them to avoid becoming trapped [56]. Further research is needed to ascertain whether this phenomenon could affect the biological control capability of the spider.

One limitation of these spiders as potential biological control agents is that their capture webs are specialised to capture arthropods that fly, jump, or fall into them, and are, therefore, unable to control pests at larval instars. The inclusion of the spider into an Integrated Pest Management (IPM) system, such as pairing it with entomopathogenic nematodes, which are already marketed as biological control agents of phytophagous larvae [17,24,57], could improve its efficacy as a biological control agent. It has been suggested that *Steinernema feltiae* nematodes may have the propensity to facilitate up to 68% mortality in larval *T. absoluta* [57]. This combination could potentially control the leafminer at both larval and adult instars, negating the shortcomings of both constituents of the management system. This approach may also reduce pesticide reliance in agricultural tomato crops, resulting in reduced pollutants in soils, waterways, and food chains [15].

## 5. Conclusions

In this study, we found that facultatively group-living *C. citricola* spiders caught leafminers and flightless fruit flies at the same rate in lab-based trials, and that larger capture web production coincides with the tomato planting and growing season in southern Spain, suggesting that this communal spider could be a potentially successful candidate for use as a biological control agent of *T. absoluta*. These findings open doors for the use of group-living arachnids to control agricultural pests, reducing commercial pesticide dependence, and having significant beneficial outcomes for environmental and economic stability, particularly in LMICs [14]. Furthermore, many LMICs exist within the overlapping geographic ranges of both *C. citricola* and *T. absoluta* [10,19,28], meaning that the introduction of pest control spiders in these regions will be unlikely to significantly damage native biodiversity. Although these results are promising, downsides to the use of spiders as pest control agents still remain; the two main issues raised being: (a) that spiders are generalist predators and are likely to catch integral tomato-pollinating arthropods [7,18,44]; and (b) that increasing spider populations will also alter community ecology, and may result in increasing predator and kleptoparasite densities [25,29,33,38]. Therefore, it is crucial that community ecology is monitored after the introduction of biological control spiders in order to preserve the health of the ecosystem and to ensure that maximum biological control efficacy is maintained. Future studies are now needed to test the efficacy of *C. citricola* for pest control in field setting, and to test the potential of other promising group-living spider species to provide pest control [44,58,59].

## Figures and Tables

**Figure 1 insects-14-00034-f001:**
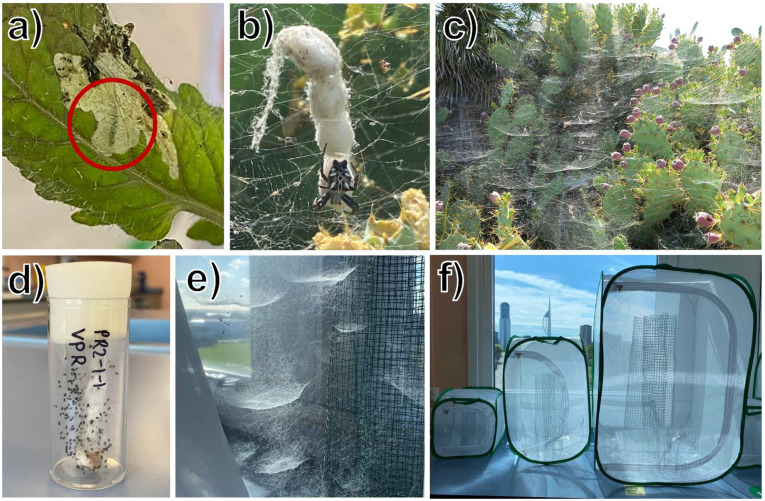
(**a**) *Tuta absoluta* larva (circled in red) feeding on the mesophyll layer of a tomato leaf. Picture taken in laboratory conditions; (**b**) an adult *C. citricola* individual in natural field settings in southern Spain, with four egg sacs; (**c**) *C. citricola* colony with visible individual horizontal web sheets on *Opuntia* sp. cactus in natural field settings in southern Spain; (**d**) spiderlings and egg sac in a 40 mL falcon tube; (**e**) colony of spiders on wire netting in a large sized mesh enclosure; (**f**) small, medium, and large sized mesh enclosures containing wire web supports for *C. citricola* colonies. Similar mesh enclosures were also used to rear *T. absoluta* moths on tomato plants. (**a**,**d**–**f**) were photographed in the laboratory in Portsmouth, UK. Photos: (**c**) LG; (**a**,**b**,**d**–**f**) TARM.

**Figure 2 insects-14-00034-f002:**
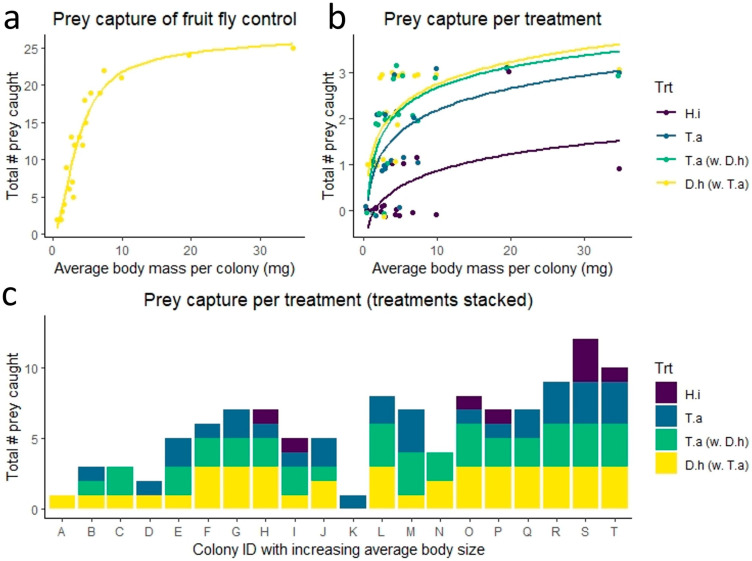
Prey capture results. (**a**) The total number of wingless fruit flies, *D. hydei*, caught after introducing five flies together, in each of five trials, in relation to the average spider body mass per colony (averaged over pre- and post-treatments weights); (**b**,**c**) The total number of prey items captured over three trials where prey was introduced singly: a black soldier fly alone (*H. illucens*) illustrated in purple; *T. absoluta* alone in blue; *T. absoluta* introduced together with a fruit fly in green; and a wingless fruit fly (*D. hydei*) introduced together with a *T. absoluta* in yellow.

**Figure 3 insects-14-00034-f003:**
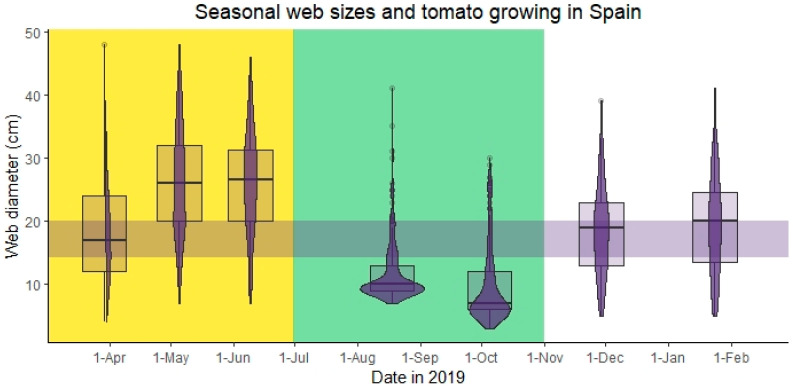
Web size seasonality and tomato growing in southern Spain. Violin plots overlaying box-and-whisker plots show the seasonal fluctuations in *C. citricola* web sizes in natural, field settings. The tomato planting and growing season is indicated in yellow (March–June) while the harvest season is indicated in green (July–Oct). A purple, horizontal band indicates the range of web sizes for which we found 100% *T. absoluta* prey capture success in our prey capture experiment in controlled lab settings.

## Data Availability

The data presented in this study are available in the Appendix A.

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
