# Peer review of "Group-Living Spider Cyrtophora citricola as a Potential Novel Biological Control Agent of the Tomato Pest Tuta absoluta"

_insects, 2022, doi:10.3390/insects14010034_

Round 1

Reviewer 1 Report

In this study, the authors explore the use of a communal living spider species, Cyrtophora citricola, as a potential biological control agent for a common moth pest in agriculture. Through a set of elegant experiments, the study concludes its usefulness providing the system can be set to control a wasp that predates on the spider. This research is novel and innovative, and potentially paradigm-shifting. Since it is also well presented and written, I have no reservations and suggest its acceptance. One suggestion the authors might consider is to weave the first paragraph from the Methods that describes the aims of the study into the end of the Introduction, as it might better find the reader's attention there. All in all, this is nice research and I am looking forward to seeing it published.

Reviewer 2 Report

Line 48 – year of describing the species Tuta absoluta is missing.

Line 60 – It should be corrected as follows: Cyrtophora citricola (Forskål, 1775) (Arachnida: Araneidae).

Line 102 – the author and the year of description of the taxa Bascillus thuringiensis is missing. In arachnological literature (taxonomy, faunistic, ecology, etc) when some taxon is mentioned for the first time in the text, it should be given in complete format: Latin name, author and year of description. But some authors or journals make exceptions of this rule.

So, according the Catalogue of Life (https://www.catalogueoflife.org/) the complete and correct name of the above mentioned species will be Bacillus thuringiensis Berliner, 1915 (not Bascillus). When a taxon is mentioned for the first time in a complete format, every subsequent time the same taxon should be written in the text only with the Latin name, without the author and year of description. For this reason, please provide the author and year of description for all taxa mentioned for the first time in the text.

Lines 171-175 – This part is very vague, imprecise and very difficult to be understood. Even more confusion is created with the data shown in the legend of fig. 1. Please use into consideration my comments on the text in the legend of fig. 1 too.

It seems that one of the main subjects of this MS, laboratory experiments on the spider C. citricola, or to be more precisely, the mesh enclosures are not properly introduced/presented in the MS. In the paragraph materials and methods only two types of mesh enclosures are given and explained: larger 1 L (12 cm diameter, 15 cm length) and large (40x40x90 cm). As the authors for these two types gives two different and not compatible metric representations, I can assume that their geometrical shapes are not compatible too. For the first one, the “larger 1 L” they give only two dimensions – diameter and length, which diameter indicate that it should be a cylinder shape mesh enclosure. For the second one, the “large“ they presented three dimensions – length x width x height, which indicate it should be a rectangular shape mesh enclosure. But on the fig. 1f, all three types of mesh enclosures are rectangular shape only! Very confusing and incomprehensible. For both types of enclosures, the authors used very similar words such as “large” and “larger”, which is very difficult to practically/physically separate them and understand which one is which?! Defining the “large” enclosure, authors provided even the most confusing information!! For the enclosure with dimensions 40x40x90 cm which most likely means length x width x height (enclosure is 90 cm high), authors mentioned rolled up rectangular wire inserted inside the enclosure with dimensions 70x100 cm, which most likely means width x height, resulting that rolled up rectangular wire is 100 cm or 1m high, 10 cm higher than the enclosure itself. But in fig. 1f it can be clearly seen that the dimensions (height) of the rolled up rectangular wires are significantly lower than the total height of the enclosures themselves!?

All this indicate that the experiment might be setup incorrectly, which will most certainly have a negative impact on the results of the experiment.

Line 177 –  Why complicate the text using abbreviations for months? In prey capture assays only juvenile spiders were used. But what about adult spiders? Are there some particular reason why adult spiders were not used in the experiment? Authors did not provide any information in the MS why the adult spiders were excluded from the experiment!

Line 182 – What does “CABI, Ghana” mean? Is that mean that additional T. absoluta pupae were imported from Africa, Ghana (country)? Why the authors included T. absoluta pupae from an another continent? Is there some specific reason for that? The authors should be aware that populations from central Africa, might have different ecological requirements/accommodations/behavior in contrast to population used in this MS, originated from UK, with much different/colder climate in contrast to tropical Africa. That would have to be very well explained in the MS.

Lines 185-188 – There is information that the larvae and tomato plants were contained in mesh cages of varying sizes – small, medium and large presented on fig. 1f. But in the legend of fig. 1f is mentioned that these three types of enclosures were for the spider C. citricola colonies only! Is that logically mean that larvae and tomato plants were contained in the same enclosure together with the spiders? If not, in that case you cannot use the same photo, in this case fig. 1f in both cases and please rewrite these sections in the MS and make the text to be clear and precise! It is recommended additional photos of larvae and tomato plants enclosures to be presented into the text.

Line 215 – This sentence is unclear, very confusing and in contradiction with the data provided in the paragraph materials and methods (line 177) where it is clearly stated that all the spiders used in prey capture assays were juveniles!!

Lines 216-218 – Again very confusing and in pure contradiction with the statement (line 177, but also line 435-436) that all the spiders used in prey capture assays were juveniles!! What is going on here! Why authors write such a contradiction! This gives the impression that they are not aligned in their views or do not read their written text carefully!

Line 219 – What is this? Small mesh enclosures are not mentioned and defined in the paragraph materials and methods!!

Lines 224-227 – The death spiders in the mesh enclosures were replaced by the authors. But in natural conditions such an artificial intervention is impossible/unfeasible to be done in such an extra quick moment. For that reason, the results obtained from this experimental study, very likely might be not correct/overestimated in contrast to results obtained from real/natural spider populations.

Line 331 – Some of the legends on fig. 1 are imprecise and please complete them with relevant information which you can find in the following question. Please be more concise and precise in your explanation:

a) Origin of the larva of Tuta absoluta presented on fig. 1a is missing. Was that larva photographed under laboratory conditions, in its natural habitat in southern Spain or in some other place?

b) and c) Origin of the exact geographical location on the spider and the webs presented on fig. 1b and 1c respectively are missing. Please provide the exact locations.

d) The falcon tube is photographed in the lab in UK, in southern Spain or in home conditions of some of the authors?

f) On fig. 1f authors mentioned three types of mesh enclosures (small, medium and large) while in the paragraph materials and methods only two types are given and explained: larger 1 L (12 cm diameter, 15 cm length) and large (40x40x90 cm)?! It remains completely unclear to me, which one is the medium and which one is the large? In the larger 1 L authors mentioned only two dimension 12 cm diameter and 15 cm length. But what about the high? Even more confusing, what does the abbreviation “1 L” mean?! The strangest and very confusing, 12 cm diameter of what? Diameter as measurement is using when dealing with circle objects, conus, cylinder etc. But on fig 1f I can see only rectangular shapes of the mesh enclosures! As the small mesh enclosures it seems are not mentioned in materials and methods, it is safe to me to conclude that they were never used in this study?

Lines 339-347 – I am unable to understand this.

Lines 429-431 – Authors stated that “…prey capture potential of a spider colony could potentially be predicted by estimating average web sizes”. But they did not give explanation how this prediction could be realized.

Lines 507-515 – The observation that multiple instances of T. absoluta were uncaptured in C. citricola webs in all colonies is crucial for the experiment! It is not really clear why the authors did not investigate this phenomenon in manner to try to find reasons for that.

Reviewer 3 Report

The manuscript addresses a topic of great interest about the role of a spider species in the control of a tomato pest. The authors focused their study on a communal spider species for which there are no studies that refer to their role as biocontrollers. Furthermore they raise the advantages that communal webs have in relation to solitary species. A species of wide worldwide distribution was selected, for which it can be potentially used in various parts of the world for biocontrol. The design of the experiences and the analyzes used are very well designed and appropriate to the objectives. The results are relevant in that the group living C. citricola has  the ability for capture Tuta absoluta and the larger capture web production coincides with the growing season of the crop. Also the authors demostrated that body size had a positive correlation  with the web size, so it could be considerated a good predictor for success predation.  This study opens new paths for the development of lines of work on biocontrol using species of communal spiders.

I suggest that with the following minor changes the manuscript is ready to be accepted for publication:

1. The aims of the study should be moved to the Introduction section (Lines 151 to 160). 

2. Line 198. Hermetia illucens could be indicated as H. illucens and Drosophila hydei as D. hydei.

3. The reference of figures 1d-f are missing from the text.

4. Line 545. For future studies it could be considered the relation of the tridimentional architecture on the capture eficiency. An article that could be cited in relation to that is the following:

Isabelle Su, Markus J. Buehler. 2020. Mesomechanics of a three-dimensional spider web, Journal of the Mechanics and Physics of Solids, Volume 144, https://doi.org/10.1016/j.jmps.2020.104096.

Round 2

Reviewer 2 Report

Thanks to the authors for incorporating almost all my comments and remarks in the MS. The previous problematic parts of the text are now much clearer.